# Stratosphere–Troposphere Exchange and Surface Ozone Pollution over Tropical Regions: A Case Study of Rossby Wave Breaking and Tropopause Folding

Clemente Lopez-Bravo<sup>1,2</sup>, Ernesto Caetano<sup>3</sup>, and Armenia Franco-Díaz<sup>4</sup>

**Correspondence:** Clemente Lopez-Bravo (c.lopez\_bravo@unsw.edu.au)

Abstract. Stratosphere–troposphere exchange (STE) is a key process by which ozone-rich stratospheric air enters the troposphere, influencing surface air quality. This study analyses an atypical STE event over North America between 6 and 14 March 2016, coinciding with a Phase I ozone contingency in Mexico City. Using ERA5 reanalysis, potential vorticity (PV) diagnostics, ozone tracers, Lagrangian trajectories, and isentropic analyses, the event is linked to anticyclonic Rossby wave breaking, a cut-off low, and a persistent tropopause fold. Deep intrusions of high-PV air reached mid- and lower-tropospheric levels, with maximum downward transport one day before the contingency. Equatorward wave amplification enabled coherent isentropic transport, allowing ozone-rich air to descend efficiently over elevated basins in Mexico. Backward trajectories confirmed stratospheric origins, while isentropic advection quantified quasi-horizontal transport along 320–340 K surfaces. Tropopause folding, strengthened by the subtropical jet and local topography, contributed an ozone mixing ratio of 8 × 10<sup>-8</sup> kg kg<sup>-1</sup> near the surface, acting as a precursor to exceedance levels. The study also identifies recurrent tropopause folds preceding highozone episodes, underscoring the recurring influence of STE on regional air quality. These findings highlight how topography, Rossby wave dynamics, and quasi-horizontal transport pathways modulate surface ozone at low tropical latitudes. These emphasize the importance of monitoring synoptic precursors and incorporating STE diagnostics into high-resolution air quality forecasts to improve prediction in complex subtropical environments. This case demonstrates how mid-latitude disturbances can directly affect tropical air quality during boreal winter–spring.

## 1 Introduction

Stratosphere-troposphere exchange (STE) is a crucial atmospheric process that transports air masses, including trace gases like ozone (Bates and Jacob, 2020; Holton et al., 1995; Martin, 1998; Wang and Fu, 2021), across the dynamic boundary that separates the stratosphere and troposphere, known as the tropopause. Tropopause folds are recognized as a primary mechanism for STE, which is often associated with upper-tropospheric jet streams and mid-latitude baroclinic systems. These tropopause folds facilitate the downward transport of stratospheric ozone into the upper and middle troposphere (Škerlak et al., 2014, 2015; Sprenger et al., 2003). However, this mechanism has been observed over subtropical and tropical areas during the boreal winter

<sup>&</sup>lt;sup>1</sup>Climate Change Research Centre, University of New South Wales, Sydney, Australia.

<sup>&</sup>lt;sup>2</sup>ARC Centre of Excellence for Climate Extremes, Sydney, Australia

<sup>&</sup>lt;sup>3</sup>Instituto de Geografía, National Autonomous University of Mexico, Coyoacan, Mexico

<sup>&</sup>lt;sup>4</sup>Karlsruhe Institute of Technology, Karlsruhe, Germany

(Barrett et al., 2019; Ogino et al., 2013; Toihir et al., 2018). Tropopause folds are frequently linked to Rossby wave breaking (RWB) events, where large-amplitude wave disturbances on the tropopause surface evolve into overturning structures, inducing potential vorticity (PV) anomalies and enhancing cross-tropopause mixing (Gabriel and Peters, 2008; Luo et al., 2019; Postel and Hitchman, 1999), sometimes exacerbating the already low air quality in cities and megacities around the globe (Chen et al., 2024; Li et al., 2025; Ni et al., 2024; Vazquez Santiago et al., 2024).

Although stratospheric intrusions are often thought of as large scale or remote phenomena, the effects of these intrusions have important local implications for surface air quality. Ozone originates both from photochemical production and from stratospheric sources (Grewe, 2006). Stratospheric ozone transported into the lower troposphere can contribute significantly to background ozone levels and, in specific meteorological conditions, even to surface exceedances of air quality standards (Knowland et al., 2017; Lin et al., 2015; Nguyen et al., 2022; Wang and Fu, 2021).

Mexico city is a megacity located at a latitude of approximately 19°N and an elevation of approximately 2,300 meters above sea level in a basin surrounded by mountains. Climatological evidence suggests that these geographical conditions make the region particularly susceptible to mid-latitude dynamical processes during the boreal winter (Binder and Wernli, 2025; Boothe and Homeyer, 2017; Škerlak et al., 2014). The southerly extension of the mid-latitude waveguide in this season allows baroclinic systems and upper-level troughs to reach subtropical and sometimes tropical latitudes (Hoskins and Karoly, 1981; Karoly and Hoskins, 1983; Nakamura, 1992; Trenberth, 1991), including central Mexico. The location of Mexico City also favours stable and dry atmospheric conditions during boreal winter, significantly contributing to the concentration of pollution emission into the local atmosphere, triggering episodes of high concentration of ozone near the surface. In addition, significant pollution emissions can produce an environment in which atmospheric processes contribute to increase near-surface ozone concentrations before high-concentration episodes occur. However, detailed case studies linking stratospheric intrusions with documented high-ozone episodes in Mexico City are limited.

Barrett et al. (2019) presented findings from surface observations and large-scale analyses on a tropopause fold over Mexico City. Although that study highlighted the important role of STE in driving extreme surface ozone concentrations, the dynamics of STE to understand the contribution of stratospheric ozone to near-surface levels remain largely unexplored. In addition, there is a lack of detailed information regarding the structure and evolution of the synoptic system over Mexico, especially from a Lagrangian and isentropic perspective. Our present study addresses these gaps by combining reanalysis data with Lagrangian trajectory and isentropic analyses to characterize both the STE event and its impact on surface ozone.

An unusual STE event occurred over Mexico between 6 and 14 March 2016. This event was driven by a mid-latitude cyclone and characterized a RWB and deep tropopause folding, which was associated with a significant increase in near-surface ozone concentrations in Mexico City and other Mexican urban centers. An analysis of historical environmental reports on air pollution episodes by local authorities shows that this episode is significant as one of the major high-concentration ozone events in Mexico City (DMA, 2025).

This study provides a dynamical analysis of the March 2016 event, focusing on the structure of the Rossby wave breaking, the evolution of the tropopause fold, and the spatiotemporal patterns of ozone tracers over Mexico City and other urban regions of Mexico. By combining reanalysis data, isentropic analysis, Lagrangian trajectories and satellite products, we show that this

large-scale STE event had measurable and regionally significant impacts on surface ozone, particularly in Mexico City, where it likely acted as a precursor to a multi-day high-ozone pollution episode.

Our findings underscore the importance of considering large-scale stratospheric processes in the forecast and management of air quality in tropical and subtropical urban regions. Given the expected urban expansion and the frequency of ozone exceedance days in Mexican cities, monitoring RWB and associated tropopause dynamics could enhance early warning systems for pollution events during boreal winter.

This paper is organized as follows: Section 2 describes the study period, the selection of the March 2016 STE event, and the datasets and methodology used, including reanalysis data and Lagrangian isentropic analysis. In Section 3, we present a dynamical analysis of the event, which covers the structure of the Rossby wave breaking, the evolution of the tropopause fold, and the spatiotemporal distribution of stratospheric ozone tracers over Mexico City and other urban regions. Finally, Sections 4 and 5 provide a discussion and conclusions, respectively.

## 1.1 Case Overview

Between 06 and 14 March 2016, a deep upper-tropospheric trough associated with a mid-latitude cyclone propagated equatorwards into central Mexico, culminating in a stratospheric intrusion over Mexico City characterized by a distinct tropopause fold (Fig. 1). Analysis of the European Centre for Medium-Range Weather Forecasts (ECMWF) Re-Analysis version 5 (ERA5; Hersbach et al., 2020) fields reveals the baroclinic nature of this system: the surface low, identified as a local geopotential height minimum at lower levels, and the upper-level trough, represented by the maximum relative vorticity at 300 hPa, which exhibit a consistent westward tilt with height, in agreement with classical baroclinic theory (Holton et al., 1995; Hoskins and Karoly, 1981). To prove this, we analyze a sequence of days between 8 and 10 March 2016. On 8 March, the surface low was located near 29.75°N, 102.25°W (682.3 m geopotential height), while the 300 hPa trough was cantered at 24.75°N, 106.75°W (vorticity maximum 2.87×10<sup>-4</sup> s<sup>-1</sup>), corresponding to a horizontal displacement of 711 km along an azimuth of -140° from the surface low towards the upper trough. By 9–10 March, the low and upper trough evolved poleward and eastwards, maintaining the westward tilt with height (displacements of 1170 km at -72.5° on 9 March and 1028 km at -96.9° on 10 March), consistent with the progressive development of a baroclinic wave. The temporal tendency of the surface geopotential height indicates slight deepening on 8 March (-1×10<sup>-4</sup> m s<sup>-1</sup>), followed by minor changes on subsequent days, highlighting a dynamically coherent mid-latitude system driving the equatorward intrusion. These diagnostics demonstrate that the extratropical cyclone satisfies the theoretical framework of baroclinic development, with the observed tilt between low-level and upper-level features acting as a key mechanism in steering stratospheric air into the subtropics (Hoskins, 1997; Masato et al., 2012)(B. Hoskins, 1997; Masato et al., 2012).

This stratospheric intrusion preceded a high ozone episode in Mexico City between 12 and 14 March 2016 (Fig. 2). Ground-based air quality records show that the event triggered an environmental contingency (monitoring locations in Fig. S1 in the supplementary material), as near-surface ozone concentrations exceeded Mexico City's threshold, reaching 159 IMECA (Metropolitan Index of Air Quality; Ezcurra, 1991). This corresponds to maintained ozone levels above 100 parts per billion (ppb) for nearly five consecutive days from 12 to 17 March 2016 (Fig. 2b). Similar ozone anomalies were observed across the

Figure 1. Relative vorticity (shading,  $\times 10^{-5}$  s<sup>-1</sup>) at 300 hPa at the time when the cyclone reached its minimum sea level pressure over Mexico (03:00 UTC, 09 March 2016). Solid black lines indicate geopotential height (m).

monitoring networks of Toluca City (Fig. 2c), as well as in western Mexico (Jalisco; Fig. 2a), indicating a regional-scale event. A well-defined tropopause fold developed during the two days preceding the ozone peak in Mexico City (light blue shading in Fig. 2), suggesting that synoptic-scale dynamics contributed to elevated local ozone concentrations. The tropopause fold was identified in the ERA5 reanalysis data by combining regions with PV < 1 PVU (1 PVU =  $10^{-6}$  K m<sup>2</sup> kg<sup>-1</sup> s<sup>-1</sup>) and enhanced ozone mixing ratios, indicating favorable conditions for STE, which is analyzed in this study.

# 2 Data and Methods

# 2.1 Reanalysis and Observational Datasets

The ERA5 global reanalysis was used at a horizontal resolution of  $0.25^{\circ} \times 0.25^{\circ}$  with hourly output and vertical resolution of 37 levels. ERA5 provides fields of PV, wind, temperature, and specific humidity, which are essential for the analysis of the dynamical evolution of the upper troposphere and lower stratosphere during the STE event over Mexico in March 2016. The primary variables derived from ERA5 include: (1) PV on isentropic levels, used to identify stratospheric air masses and tropopause folds; (2) wind speed and geopotential height at constant pressure levels to characterize the synoptic-scale evolution of the Rossby wave breaking and the associated cut-off low event over Mexico, and (3) ozone mass mixing ratio from ERA5 at constant pressure for the analysis of the downward transport of stratospheric ozone into the troposphere during the STE event.

**Figure 2.** Observed and reanalysis evidence of the high-ozone event in Mexico during 12–14 March 2016. (a) Ground-based ozone concentrations from the air quality monitoring network in western Mexico, Guadalajara, Jalisco, (b) Mexico City, and (c) Toluca (Locations are shown in Fig. S1). Three locations show sustained near-surface ozone concentrations (ppb). Light blue shading denotes the tropopause fold identified from ERA5 reanalysis using potential vorticity values 

120

125

## 2.2 Lagrangian trajectories

Lagrangian trajectories were computed using the Lagrangian Analysis Tool LAGRANTO (Sprenger and Wernli, 2015; Wernli and Davies, 1997) to investigate STE over Mexico. LAGRANTO allows flexible and precise definition of trajectory starting positions based on geometrical or meteorological criteria. For this study, the trajectories were initialized equidistantly (50 km spacing) over a box covering Mexico City and its Metropolitan area (Longitude: 100°W to 98°W, Latitude: 18.5°N to 20°N) and on multiple vertical levels between 700 and 100 hPa, with the 700 hPa level chosen to account for the altitude of Mexico City (~2,300 m above sea level). Backward trajectories were calculated for 72 hours using ERA5 reanalysis meteorological fields as input. The trajectories were then selected based on dynamical criteria, specifically potential vorticity (PV > 2 PVU), to identify stratospheric air masses linked to the Rossby wave breaking and the cut-off low event over Mexico. The use of LAGRANTO enabled a detailed investigation of the downward intrusion of high-PV air along tropopause folds and its connection to the cut-off system over Mexico.

# 2.3 Rossby wave breaking identification

A RWB event was identified over Mexico in March 2016 using the Python package WaveBreaking (Kaderli, 2023), designed for detecting, classifying, and tracking RWB in atmospheric fields. The detection is based on analyzing the dynamical tropopause, represented by closed PV contours (e.g., the 2 PVU line), and applying three RWB indices. The Streamer Index identifies filamentous structures (streamers) in the tropopause contour that are geographically close but distant along the contour path (Sprenger et al., 2017; Wernli and Sprenger, 2007). The Overturning Index detects the overturning of the contour, defined as at least three intersections of the contour within the same longitude (Barnes and Hartmann, 2012). The Cut-off Index characterizes the decay of a wave breaking event by identifying streamers that separate from the main stratospheric or tropospheric body (Wernli and Sprenger, 2007). The algorithm calculates RWB characteristics such as area, intensity, and temporal evolution, allowing a systematic analysis of Rossby wave breaking and its connection to STE over Mexico.

### 2.4 Isentropic Analysis of Tropopause Fold Transport

To quantify the downward transport of stratospheric air over Mexico City, we calculated transport across the dynamical tropopause using ERA5 winds, PV, and ozone interpolated onto isentropic surfaces following Dethof et al. (2000). Air parcels with PV > 2 PVU were tracked to identify regions of tropopause erosion, and negative advection values were interpreted as indicative of the export of high-PV stratospheric air into the troposphere. Ozone mixing ratios were also used to further distinguish stratospheric signatures. This approach allows the detection of quasi-horizontal and irreversible transport along isentropic surfaces (Dethof et al., 2000), providing a framework to relate stratospheric intrusions to enhancements of near-surface ozone during the 8–15 March 2016 tropopause fold event.

# 135 2.5 Tropopause Definition and Stratospheric Intrusion Diagnosis

Rossby waves arise from the latitudinal variation of the Coriolis parameter and the conservation of absolute vorticity (Holton, 1997). Rossby wave breaking (RWB) occurs when the wave amplitude grows to the point where the crest or trough overturns. It is characterized by the rapid, irreversible deformation of PV contours on isentropic surfaces and is commonly diagnosed by a reversal of the meridional PV gradient (Postel and Hitchman, 1999). The RWB also implies the disruption of the westerly flow in the neighboring areas (Tyrlis and Hoskins, 2008) and it promotes blocking.

The dynamical tropopause is commonly identified on isentropic surfaces as the potential vorticity (PV) contour coinciding with the strongest meridional gradient, which acts as a barrier to cross-tropopause transport (Holton, 1997; Holton et al., 1995). Typical PV thresholds range from 1.5 to 5 PVU (Jing and Banerjee, 2018; Schoeberl, 2004), with the value associated with the maximum gradient generally increasing with potential temperature (Homeyer and Bowman, 2013; Turhal et al., 2024). In extratropical and subtropical studies, the 2 PVU surface is widely used to separate stratospheric and tropospheric air masses and to identify tropopause folds (Holton et al., 1995; Sprenger et al., 2003; Tyrlis and Hoskins, 2008). Tropopause folds occur where the 2 PVU surface dips below 400 hPa and coincides with regions of large PV gradients and relative vorticity anomalies, marking zones of potential cross-tropopause exchange.

In this study, we employ two complementary approaches to examine and facilitate the interpretation of RWB and tropopause fold over Mexico. First, we define the tropopause as a surface of constant potential vorticity (PV). Second, we use the isentropic tropopause definition to analyze its variability along potential temperature surfaces. We apply PV thresholds of 2, 3, 4, and 5 PVU to define the tropopause on the 320 K and 380 K at 10 K.

To illustrate these concepts in the context of the March 2016 event, Figure 3 presents both the monthly mean PV- $\theta$  structure over North America (Fig. 3a) and the case-specific tropopause features over Mexico. Figure 3b shows the corresponding case study at 21 UTC 10 March 2016 over Mexico, where a PV maximum is evident over the extratropic near 20°N, and values between 1 and 4 PVU extend below 300 hPa (see red box in Fig 3b). These regions are characterized by sharp gradients and downward structures of the 2 PVU surface. Features are tracked to assess the depth and progression of the intrusion into the lower troposphere over central Mexico. For the second approach, we analyze isentropic surfaces, as illustrated by the animation of wind streamlines at 315 K, 330 K, and 340 K (Animation 1), which shows that air primarily follows these surfaces.

## 160 3 Results

# 3.1 Synoptic-scale evolution of the March 2016 event

Between 6 and 14 March 2016, a pronounced upper-tropospheric trough deepened over the eastern North Pacific and propagated equatorwards into central Mexico, associated with a mid-latitude cyclone. By 8 March, the trough extended into the subtropical and tropical latitudes, enhancing meridional transport and reversing the meridional PV gradient, features that are characteristic of anticyclonic Rossby wave breaking (AWB) (Gabriel and Peters, 2008; McIntyre and Palmer, 1983; Tyrlis and Hoskins, 2008). The evolution of the AWB continued, and by 10 March the system had amplified into a cut-off low centered

Figure 3. Zonal mean potential temperature (black contours, K) and potential vorticity (colour shading, PVU, 1 PVU =  $1.0 \times 10^{-6}$  K m<sup>2</sup> kg<sup>-1</sup> s<sup>-1</sup>) from ERA5. (a) March 2016 monthly mean over North America. (b) 21 UTC 10 March 2016 over Mexico, showing a PV maximum over the extratropic ( $\sim 20^{\circ}$ N) and downward extension of PV values between 1 and 4 PVU below 300 hPa, indicative of tropopause folding and potential cross-tropopause exchange (red box highlights region of interest).

near 20°N, coinciding with the southern displacement of the subtropical waveguide during boreal winter. This synoptic configuration favored strong advection of high-PV, stratospheric air into the tropic and subtropics regions, setting the environment for a STE event. Analyses with ERA5 data indicate that the trough development was dynamically coupled with the subtropical jet, with wind speeds at 300 hPa exceeding 50 m s $^{-1}$  along the jet entrance region (Fig. 4), consistent with conditions

180

200

known to promote RWB. Meridional cross-sections of wind show the jet stream positioned near 20°N, indicating a pronounced equatorward displacement relative to its climatological latitude (not shown).

Contours that experienced RWB were identified on each isentropic surface. Figure 4 shows PV contours from 2 to 6 PVU on the 330 K isentropic surface. The contours overturn across North America, the Pacific Ocean, and the Atlantic Ocean, with PV values increasing meridionally from the equator to the pole, consistent with a standing Rossby wave. We define overturning as the point where a PV contour intersects a meridian, as illustrated in Fig. 4a over Mexico and the United States. The overturning also satisfies the 1,000 km distance criterion used to identify the structure as part of the same RWB event (Jing and Banerjee, 2018). This event is classified as anticyclonic Rossby wave breaking, characterized by cyclonically wrapping PV contours and clear signs of equatorward amplification over Mexico. Similar patterns are observed on the 320–350 K isentropic surfaces (not shown).

During 9–11 March 2016 (Figs. 4a–d), the 2–6 PVU contours show structures of equatorward amplification. The amplification of the isentropic PV field reflects the Rossby wave amplification process, which drives a dynamical tropopause folding over Mexico City. The development of low-pressure systems associated with tropopause folding follows the theoretical basis described by Hoskins et al. (1985). As stable stratospheric air descends into the troposphere, air parcels acquire cyclonic vorticity (Hoskins et al., 1985), which can result in a cold-cored, closed cyclonic circulation forming of a cut-off low system, moving over Mexico. The intrusion of highly stable, high-PV, stratospheric air strengthens cyclonic motion aloft and extends this influence downwards, particularly over the mountain ranges of central Mexico (elevations >2000 m above mean sea level). This process intensifies on 10–12 March 2016 as the intrusion deepens toward the surface, as discussed later.

# 3.2 Rossby wave breaking characteristics

Objective RWB diagnostics confirm the presence of a large streamer and subsequent cut-off formation between 8 and 11 March. The streamer extended southwards from mid-latitudes into central Mexico, reaching latitudes below 20°N, and eventually evolved into a cut-off low. The Overturning Index showed multiple crossings of the 2 PVU contour along different longitudes, indicating a deep and persistent regime of RWB. This coherence lasted nearly three days, which is longer than the typical duration of 1 to 2 days for most low latitude intrusions. As a result, there was sustained cross-tropopause exchange over central Mexico.

The dynamical structure of the synoptic system of this event is explored combining isentropic PV analysis at 320 K with geopotential height at 500 hPa. Figure 5 shows the streamer and a cut-off low extending over Mexico. The STE occurs when the tropopause is strongly meandering by large latitudinal displacements. The streamer is manifested as a tongue of anomalously high-PV air extending equatorwards (Fig. 5a), generated by the large-scale deformation field acting on the wave structure. Some parts of this streamer experience filamentation, while others form an isolated, coherent structure containing high-PV air, resulting in a cut-off cyclone (Figs. 5b and 5c). The collocation of the PV at 320 K streamer with a closed 500 hPa geopotential contour confirms the dynamical coupling between upper-level PV and mid-tropospheric cyclonic circulation (Figs. 5c and 5d). Cut-off cyclones of this type have been argued to play an important role in STE processes (Holton et al., 1995), as these structures can promote the irreversible descent of stratospheric air into the troposphere. In this case, the development of the

**Figure 4.** PV contours (2–6 PVU) on the 330 K isentropic surface and shaded 300 hPa wind speed (m s<sup>-1</sup>) from ERA5 during 9–11 March 2016. Contours undergoing Rossby wave breaking (RWB) overturn across North America, the Pacific Ocean, and the Atlantic Ocean. Overturning is defined where a PV contour intersects a meridian and spans at least 1,000 km. Cyclonic RWB is evident over Mexico, with contours wrapping cyclonically and showing equatorward amplification, corresponding to tropopause folding and the development of low-pressure systems. Panels (a)–(d) show the temporal evolution of this event leading to a cut-off low event over Mexico.

cut-off low over central Mexico exemplifies the dynamical pathway from large-scale Rossby wave meander to localized STE, reinforcing the link between isentropic PV diagnostics and mid-tropospheric circulation features.

# 3.3 Tropopause fold depth and structure

The dynamical tropopause is a line of constant PV that separates the highly stratified stratosphere from the weakly stratified troposphere (Morgan and Nielsen-Gammon, 1998). It is identified using PV thresholds ranging from 1 to 5 PVU. Between 9 and 11 March, analyses reveal the development of a well-defined tropopause fold associated with the upper-level trough. The 2 PVU surface descended to altitudes below 400 hPa, a clear indicator of downward penetration of stratospheric air (Figs. 3b). Vertical cross-section at 00 UTC 12 March 2016 shows sharp PV gradients and a folded structure of the tropopause extending over central Mexico and Mexico City (Fig. 6), persisting for ~24 hours prior to the onset of high ozone surface concentrations on 13 March 2016.

Figure 6 illustrates the dynamical structure of this fold, where a tight PV gradient is collocated with a pronounced increase in the squared Brunt-Väisälä frequency ( $N^2$ ), marking the tropopause. The tropopause folding exhibited high-PV air descending into low-PV air, locally reduced  $N^2$  within the intrusion (weakened stability), and strong  $N^2$  above, consistent with a stratospheric intrusion extending into the mid-lower troposphere below 300 hPa and latitude  $\sim 20^{\circ}N$ . A specific humidity threshold

**Figure 5.** Isentropic potential vorticity in (shading, PVU) on the 320 K surface overlaid with 500 hPa geopotential (thin contours at 50 intervals, m<sup>2</sup> s<sup>-2</sup>) for (a) 09 UTC 09 March 2016, (b) 06 UTC 10 March 2016, (c) 16 UTC 10 March 2016 and (d) 09 UTC 11 March 2016. A streamer of high-PV air extends equatorwards over Mexico (thick black contour), indicating strong tropopause distortion. Portions of this streamer roll up into an isolated structure, forming a cut-off low collocated with a closed 500 hPa circulation (red contours). This configuration highlights the dynamical coupling between upper-level PV and mid-tropospheric cyclonic development, a key mechanism of STE exchange for this case study.

of q = 0.1 g kg<sup>-1</sup> was also applied to identify stratospheric air from diabatically generated high-PV values in the troposphere (solid red line in Fig. 6). This combination of PV,  $N^2$ , and humidity diagnostics provides a comprehensive framework for identifying STE during the March 2016 cut-off low event over low latitudes and mountain ranges.

The tropopause fold developed on the central-southern flank of a low-pressure system along the westerly jet stream. Three regions of folding were identified: over the United States ( $\sim$ 40°N and  $\sim$ 30°N) and over Mexico ( $\sim$ 20°N), associated with

**Figure 6.** Tropopause fold over Mexico at 00 UTC 12 March 2016. The dynamical tropopause is defined using PV (solid thick black contours, PVU), with the cyan solid line indicating 2 PVU, and the squared Brunt–Väisälä frequency (colour shading,  $N^2$ , ×  $10^{-4}$  s<sup>-2</sup>). Section was average over longitudes between 99.8°W and 98.5°W and displays latitudes from 10°N to 45°N. Tropopause folds extend from the upper to mid–lower troposphere, with high-PV stratospheric air descending into low-PV tropospheric air and locally reduced  $N^2$  within the intrusion. The solid red line indicates the specific humidity threshold (q = 0.1 g kg<sup>-1</sup>). Thin solid lines indicate the potential temperature (K). The dotted blue lines display the wind speed (m s<sup>-1</sup>): 30 m s<sup>-1</sup>, 32 m s<sup>-1</sup>, 35 m s<sup>-1</sup> and 40 m s<sup>-1</sup>.

the cut-off low centered between 15°N and 25°N (Fig. 6). These regions feature a horizontally extensive, statically stable layer in the mid-troposphere, marked by a vorticity maximum induced by the cut-off low, which favours displacement of the system towards lower latitudes. Amplification of the isentropic PV contours illustrates the Rossby wave amplification process and the associated tropopause folding. In these regions, an upper-level trough develops (Fig. 1), supported by the intrusion of highly stable, high-PV stratospheric air that strengthens cyclonic circulation aloft and extends downwards to induce surface cyclonic motion. As the intrusion deepens, this process intensifies, consistent with the amplification patterns shown in Fig. 4. The concurrent amplification of the 1 PVU contour highlights the downward intrusion of high-PV stratospheric air, reflecting STE linked to baroclinic instability and Rossby wave amplification. The intrusion deepened further between 10 and 12 March 2016, producing a closed circulation throughout the troposphere (Fig. 4). At the surface, a low-pressure system first formed and then intensified during the day, coinciding with an amplifying trough in the mid- and upper troposphere.

# 3.4 Transport pathways

Isentropic analyses on the 320–340 K surfaces reveal pronounced equatorward advection of high-PV air from the mid-latitudes into central Mexico, consistent with the presence of a tropopause fold. Backward trajectories computed with LAGRANTO using ERA5 input fields confirm this intrusion. Parcels arriving near 700 hPa over Mexico City between 11 and 13 March 2016 originated from stratospheric levels above 250 hPa within the fold region. The three-dimensional trajectories capture the filamentation of a streamer, the subsequent development of a cut-off low, and the persistence of the fold, which together provided an efficient dynamical pathway for the downward transport of stratospheric air. This mechanism enabled ozone-rich, high-PV air to penetrate the lower troposphere above Mexico City (Fig. 7). Although ERA5 ozone tracer fields indicate enhanced ozone mixing ratios ( $\sim$ 8 × 10<sup>-8</sup> kg kg<sup>-1</sup>), the dominant signal was dynamical: a sustained Rossby wave breaking episode that facilitated STE at unusually low latitudes.

Ozone mixing ratios were interpolated along each trajectory, and only those with pressure differences (ΔP) exceeding 200 hPa between the initial and final positions were retained to diagnose descent trajectories. Most trajectories were located near 600 hPa at 12 UTC 12 March (Fig. 7a), with only a few extending down to 700 hPa above Mexico City (Fig. 7b) (surface pressure reference for Mexico City: 760 hPa). These parcels carried elevated ozone tracer values from the upper stratosphere that decreased gradually towards the surface, in agreement with the PV distribution showing 1–2 PVU air below 550 hPa (Fig. S2). By 00 UTC 13 March, the maximum density of descending trajectories shifted downward from 600 hPa to 700 hPa (Figs. 7c, 7d), coinciding with the observed peak in surface ozone concentrations in Mexico City (Fig. 2b).

Two distinct groups of trajectories can be distinguished at 700 hPa (Fig. 7d). The first group experienced a rapid vertical descent with a clear  $\Delta P$  from 350 hPa to 700 hPa within 48 hours, but with relatively low PV values ( $\sim$ 1 PVU) near the surface (Fig. S2d), suggesting substantial dilution during transport, e.g., mixing with tropospheric air that reduces the original stratospheric PV. In contrast, the trajectories of the second group decreased more gradually during the first 48 hours, with PV decreasing smoothly from approximately 2 PVU aloft to about 1 PVU at 700 hPa. This gradual pathway provided a more efficient channel for transporting ozone-rich air into the lower troposphere, consistent with the enhanced surface ozone levels observed during the event.

The spatial structure of the backward trajectories further illustrates the large-scale dynamical setting of the event. Parcels initially located in the stratosphere, with PV values exceeding 2 PVU, were traced back to the eastern Pacific and North America (Fig. S3), linking the fold to the Rossby wave breaking and the cut-off low embedded in the extratropical circulation. Their subsequent descent into the 300–700 hPa layer over central Mexico reflects the sloping geometry of the tropopause fold and the adiabatic descent forced by the cyclonic circulation on the anticyclonic flank of the breaking wave. Trajectories released between 500 hPa and 600 hPa reveal the evolution of the cut-off low over Mexico between 12 and 13 March 2016 (Fig. S3), emphasizing the dynamical coupling between stratospheric PV anomalies aloft and the development of cyclonic features near the surface.

Ozone transport from the lowermost stratosphere into the troposphere occurs in connection with tropopause fold event and the connection between the tropopause fold-STE and cut-off cyclone. Figure 8 illustrates the dynamical and ozone evolution

Figure 7. Backward trajectories initialized over Mexico City and its metropolitan area during 12–13 March 2016, computed with LA-GRANTO using ERA5 fields. Panels (a–b) show trajectories arriving at 600 hPa and 700 hPa at 12 UTC 12 March, while panels (c–d) illustrate the evolution at 00 UTC 13 March at the same levels. Ozone mixing ratios ( $\times 10^{-8}$  kg kg<sup>-1</sup>) are shown as shading along the trajectories. Only trajectories with  $\Delta P > 200$  hPa and PV > 2 PVU at upper levels were retained. Most parcels originated in the stratosphere above 250 hPa within the tropopause fold region, confirming stratosphere-to-troposphere transport.

of the STE event over Mexico City 8–16 March 2016. Tropopause folding is linked to upper level baroclinic waves and jet streams (Fig. 6). As the baroclinic wave develops near the tropopause, strong subsidence occurs over Mexico, accompanied by a downward tilt of the isentropes, facilitating the intrusion of stratospheric air into the troposphere between 10 and 13 March 2016, before the environmental contingency in Mexico City. This intrusion provides a pathway for stratospheric constituents, including ozone, to descend into the lower troposphere over Mexico City.

Coinciding with the tropopause fold, ERA5 ozone tracer values reveal elevated ozone mixing ratios of  $\sim 8x10^{-7}$  kg kg<sup>-1</sup> between 150 and 400 hPa on 11 March 2016 (Fig. 8), indicating significant enhancement of ozone in the mid-troposphere over tropical latitudes. The spatial and temporal alignment between ozone mixing ratio and PV anomalies confirms the stratospheric origin of the descending air mass. PV extends from the mid-latitudes into the subtropics by 9 March, with descending motion evident from increasing PV values and downward-shifting ozone tracer (Fig. 8). This multi-day progression indicates large-

**Figure 8.** STE over Mexico City, 8–16 March 2016. ERA5 ozone mixing ratio (x10<sup>-7</sup> kg kg<sup>-1</sup>) are shown in black contours and potential vorticity (PV) in shading. Ozone mixing ratio and potential vorticity were average over latitudes from 18.5°N to 20°N and longitudes from 99.5°W to 98.5°W. Tropopause folding linked to an upper-level baroclinic wave and jet stream drives the intrusion of stratospheric air into the troposphere between 10–12 March, prior to the environmental emergency (start and end days indicated by vertical red dotted lines). This event represents an unusually strong and unusual STE episode for the tropical latitude of Mexico City.

scale subsidence linked to baroclinic wave breaking, emphasizing the role of tropopause folding in driving STE and ozone transport into the lower troposphere. These processes ultimately influence ozone concentrations over densely populated and industrial regions of Mexico (Fig. S4).

To complement the ozone tracer analysis, satellite-derived products were used to assess the evolution of the total and tropospheric ozone columns during the tropopause fold and STE event over Mexico (Fig. S5). Data from The Deep Space Climate Observatory (DSCOVR) Earth Polychromatic Imaging Camera (EPIC) were employed to generate a sequence of daily averaged ozone fields between 7 and 11 March 2016, with tropospheric ozone column and total column (Kramarova et al., 2021; Marshak et al., 2018). The EPIC total column ozone shows an enhancement across Mexico. Tropospheric ozone signatures reveal a distinct increase over eastern Mexico on 10–11 March 2016, coinciding with the development of the tropopause fold. However, the tropospheric signature is not evident over central Mexico. Although the daily averages represent specific satellite scan times, the structures are robust and supported by the Ozone Mapping and Profiler Suite (OMPS)- Suomi National Polar-Orbiting Partnership (NPP) total column data (Flynn et al., 2018), which display similar spatial patterns.

# 3.5 Downward transport over Mexico City

Quasi-horizontal isentropic advection was quantified by interpolating ERA5 winds, PV, and ozone to isentropic surfaces. This approach provides a direct measure of the horizontal transport of PV anomalies along isentropic layers. In the context of the tropopause fold event, the analysis highlights regions where stratospheric air masses are irreversibly mixed into the

troposphere. For simplicity, the analysis assumes one-way flux, although exchange can occur in both directions. Emphasis is placed on negative advection values, which correspond to the export of high-PV stratospheric air, indicating erosion of the tropopause and the downward transport of stratospheric signatures into the troposphere. The 2 PVU threshold was used to delineate the dynamical tropopause, and areas with PV > 2 PVU experiencing negative advection reveal the quasi-horizontal, irreversible transport of stratospheric parcels into the troposphere, consistent with the mechanism described by Dethof et al. (2000).

Figure 9 illustrates the detection of air parcels undergoing downward transport over Mexico City and its Metropolitan Area on isentropic surfaces between 310 K and 360 K during 8–15 March 2016. Downward-moving stratospheric parcels are identified by markers with ozone mixing ratios exceeding  $5 \times 10^{-7}$  kg kg<sup>-1</sup> and PV > 2 PVU (triangles indicate parcels descending in the stratosphere, while stars display parcels descending in the troposphere). The erosion of the tropopause is particularly evident below the 320 K isentrope (starts), marking the region where stratospheric air intruded into the troposphere. This transport was strongest between 10 and 13 March 2016, descending to shallower isentropes (~310 K) during 12–13 March. This period coinciding with the environmental contingency in Mexico City. Independent calculations of cross-tropopause transport following (Siegmund et al., 1996) confirmed a similar footprint over the region (not shown), providing a robust evidence that the tropopause fold directly contributed to the enhancement of near-surface ozone levels.

# 3.6 Observed Tropopause Folds Preceding High-Ozone Events in Mexico City

Tropopause folds over central Mexico have preceded major air quality incidents, emphasizing the role of STE as a key mechanism modulating the contribution of stratospheric ozone to near-surface ozone concentrations during boreal winter in the Mexico Megacity. For instance, four events were identified where the STE mechanism contributed to increase the surface ozone concentration in Mexico City. A shallow intrusion was linked to a pollution event occurred on 25 December 2015 (Fig. 10a), while deeper intrusions on 14 April 2007 (Fig. 10b) and 10 April 2019 (Fig. 10c) were associated with consecutive tropopause folding events that clearly indicate the effects of the STE mechanism in enhancing ozone transport prior to high concentrations over Mexico City. The event occurred on 22 February 2024, also transported ozone into the lower troposphere, with elevated mixing ratios observed between 700 hPa and 600 hPa (Fig. 10d), indicating that stratospheric air reached into the planetary boundary layer of Mexico City. These cases show that both shallow and deep tropopause folds are effective pathways for stratospheric ozone, increasing surface concentrations and contributing to severe pollution episodes over the region. Additional research is necessary to understand the frequency, intensity, and mechanisms of these events, along with their impact on local air quality extremes.

#### 4 Discussion

325

This work documents an unusual STE event over central Mexico during March 2016, characterized by anticyclonic Rossby wave breaking, a persistent tropopause fold, and quasi-horizontal isentropic transport. The intrusion was unusually deep, reaching the mid- and lower troposphere, and by 13 March, stratospheric air descended to surface levels in Mexico City, coinciding

Figure 9. Air parcels detected using quasi-horizontal isentropic advection during the tropopause fold event over Mexico City and its Metropolitan Area between 8 and 15 March 2016. Downward-moving stratospheric parcels are indicated by markers with ozone mixing ratios ( $\times 10^{-7}$  kg kg<sup>-1</sup>), triangles and starts indicating air parcels above and below of the dynamical tropopause respectively and circles displays PV (PVU) on isentropic surfaces from 310 K to 360 K. Each marker displays the daily average of all pixels detected in the 24 hours analyzed.

with the Phase I Mexican environmental contingency between 12–14 March. The associated ozone analysis indicates that this stratospheric contribution preceded and potentially amplified the surface ozone concentrations, corroborating its role as a temporal precursor to the air quality extreme.

While a precise quantification of the stratospheric contribution is not possible given the use of reanalysis data, we found a contribution of ozone mixing ratio of  $\sim 8x10^{-8}$  kg kg<sup>-1</sup>. Then, the ozone mixing ratio related to the intrusion was enough to serve as a temporary precursor to the exceedance event. The stratospheric input should therefore be viewed as an additional

Figure 10. Examples of tropopause fold events over central Mexico associated with major air quality episodes. (a) shallow intrusion on 25 December 2015, ERA5 potential vorticity (PV) and ozone mixing ratio  $(x10^{-7} \text{ kg kg}^{-1})$ , (b) consecutive deep intrusions on 14 April 2007 and (c) 10 April 2019, and (d) 22 February 2024 event. These cases illustrate how both shallow and deep folds can act as effective pathways for stratospheric ozone, enhancing surface concentrations and contributing to extreme pollution episodes. Start and end days indicated by vertical red dotted lines).

source that enhanced pre-existing anthropogenic background concentrations, driving them above the contingency threshold, rather than as the dominant mechanism of the episode. The timing of the event is also notable. The intrusion reached the lower troposphere roughly one day before the environmental contingency began, suggesting a lag between the dynamical descent of ozone-rich air and the peak surface concentrations. This suggests that the stratospheric signature likely preconditioned the lower troposphere, which was then exacerbated by local photochemical ozone creation, resulting in the observed exceedances.

The dynamical evolution of the event involved streamer development, cut-off low formation, and slow-evolving baroclinic disturbances, consistent with known midlatitude-tropical interactions that facilitate stratospheric intrusions (Appenzeller and Davies, 1992; Wernli and Sprenger, 2007). Unlike more transient STE events, the persistence of the fold and its structured isentropic pathways not only favored unusually deep descent into the boundary layer over Mexico City, enhanced by its high altitude (~2300 meters above mean sea level) and the southward position of the boreal winter subtropical jet, but also enabled long-range meridional and zonal transport of ozone-rich air. These quasi-horizontal pathways connected midlatitude source regions with multiple destinations in central and northern Mexico, demonstrating how Rossby wave breaking can act as an effective pathway for trace gases and synchronize pollution events across distant metropolitan areas.

Local emissions and meteorological conditions contribute to tropospheric ozone concentrations, while long-range stratospheric transport adds to the already high anthropogenic background. This finding aligns with previous studies on STE contributions to tropospheric ozone in North America and Europe (Lin et al., 2015; Trickl et al., 2020, 2023) but represents one of

the few documented cases in Mexico (Barrett et al., 2019). Importantly, the STE influence extended beyond Mexico City, with elevated ozone mixing ratios also detected in Guadalajara, Monterrey, San Luis Potosí, and Torreón (Fig. S4). Such regional-scale transport compounds local emissions, creating simultaneous exceedances across multiple cities and turning a dynamical event into a broader national-scale air quality hazard.

The role of topography and altitude also emerges as a key modulator of intrusion depth. Mexico City and Monterrey, situated in elevated basins, could be especially susceptible to stratospheric air moving close to ground level during tropopause folds, while regions at lower elevations might experience less intense surface effects. These findings indicate that the interaction between complex terrain and intrusion dynamics requires further investigation in the tropical region, where orography significantly influences circulation during the boreal winter and spring. This phenomenon has been discussed in other geographical areas, as noted in Chen et al. (2016), Lüthi et al. (2015), and Škerlak et al. (2019). While similar STE-related ozone enhancements have been reported in North America and Europe, fewer studies exist in tropical regions where weather is influenced from extratropical cyclones. Comparisons with the Mediterranean basin and East Asia suggest that complex orography combined with subtropical jet dynamics plays a central role in facilitating deep intrusions. In Mexico, the elevated basins of central and northern cities appear particularly susceptible, underlining the broader subtropical relevance of this case.

From a forecasting perspective, these results emphasize the value of monitoring Rossby wave activity, tropopause folding, and isentropic transport pathways as potential early-warning indicators of surface pollution episodes. Current operational forecasts and air quality management plans in Mexico primarily focus on local emissions and meteorological dispersion in the planetary boundary layer. However, forecast accuracy is likely limited by strong terrain–flow interactions and the complexities of basin circulations, which modulate how stratospheric air masses interact with the boundary layer. Incorporating synoptic dynamical precursors into high-resolution numerical weather prediction and coupled chemical transport models could enhance predictive capacity and support more proactive public health interventions, particularly in mountainous tropical and subtropical regions.

### 370 5 Conclusions

This study provides a detailed dynamical analysis of a stratosphere–troposphere exchange (STE) event over central Mexico during 6–14 March 2016, coinciding with a severe ozone pollution episode in Mexico City. Using ERA5 reanalysis, PV and ozone tracer diagnostics, Lagrangian trajectories, and isentropic analysis, we show that this event was driven by anticyclonic Rossby wave breaking and a persistent tropopause fold, facilitating the transport of stratospheric air into near-surface layers over central area of Mexico. The key findings are listed below:

1. The amplification of Rossby waves toward the equator and the subsequent formation of a streamer contributed to the development of a cut-off low over Mexico. This process created coherent isentropic transport pathways that allowed ozone-rich air from the stratosphere to efficiently descend into the lower troposphere. The intrusion of this stratospheric air reached the lower troposphere about one day before the environmental contingency began, highlighting a temporal lag between the dynamical descent of stratospheric air and the peak concentrations of surface ozone.

- 2. Persistence and depth of the tropopause fold were unusual compared to previous STE events, allowing stratospheric air to reach the boundary layer over Mexico City, unlike shallower or more transient intrusions reported in earlier studies (e.g., Barrett et al., 2019).
- 3. Dynamical coupling between PV anomalies and mid-tropospheric cyclonic circulation amplified the intrusion and sustained downward transport over multiple days, demonstrating a strong link between large-scale Rossby wave dynamics and regional air quality impacts.

Elevated ozone was observed not only over Mexico City but also in urban centers, such as Guadalajara, Monterrey, San Luis Potosí, and Torreón, showing that STE events can impact air quality regionally. Intrusion depth was strongly influenced by topography and altitude, with elevated basins like Mexico City and Monterrey more prone to near-surface stratospheric air, while lower-altitude regions were less affected. Comparisons with other subtropical regions, including the Mediterranean and East Asia, indicate that subtropical jet dynamics combined with complex orography facilitate deep intrusions, highlighting the vulnerability of central Mexican urban basins.

Limitations of this study include reliance on ERA5 reanalysis, which may underestimate the sharpness of small-scale folds and mixing layers, and the lack of direct in situ ozone measurements aloft during the event. Future work should combine high-resolution chemistry-atmospheric simulations with targeted observational campaigns to quantify stratospheric contributions and better understand the dynamical–chemical mechanisms driving regional ozone exceedances.

Overall, the March 2016 case exemplifies how synoptic-scale Rossby wave dynamics and tropopause folding can directly influence surface air quality in subtropical urban basins. Recognizing these links is essential for attributing past pollution episodes, improving operational forecasts, and informing mitigation strategies. High-resolution chemistry–climate simulations, combined with real-time monitoring of STE precursors, would improve the quantification of stratospheric contributions to surface ozone, providing actionable insights for environmental management and public health preparedness.

Data availability. ERA5 reanalysis data Hersbach et al. (2020) are publicly available from the Copernicus Climate Data Store https://cds.climate.copernicus.eu. Tropospheric ozone data from the NASA Earth Probe Total Ozone Mapping Spectrometer (EPIC) instrument Kramarova et al. (2021); Marshak et al. (2018) and Ozone Mapping and Profiler Suite (OMPS) data from the Suomi National Polar-Orbiting Partnership (Suomi NPP) satellite Flynn et al. (2018) are both available through NASA's Earthdata portal https://earthdata.nasa.gov.

Code and data availability. Lagrangian trajectories were computed using the LAGRANTO tool Sprenger and Wernli (2015), which is available upon request from the developers. The WaveBreaking detection package Kaderli (2023) is accessible at https://zenodo.org/records/14214463.

Author contributions. CLB and EC conceived the study; CLB analyzed data and wrote the manuscript draft; CLB, EC, and AFR reviewed and edited the manuscript

*Competing interests.* Authors declare that they have no competing interests related to this research.

*Acknowledgements.* This work was funded by the ARC Centre of Excellence for Climate Extremes CE170100023 for C. Lopez-Bravo. We also want to thank Michael Sprenger for his valuable feedback.

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
