# Peer review of "Stratosphere–Troposphere Exchange and Surface Ozone Pollution over Tropical Regions: A Case Study of Rossby Wave Breaking and Tropopause Folding"

_EGUsphere, 2025_

## Referee Comment (RC1)

**Stratosphere–Troposphere Exchange and Surface Ozone Pollution over Tropical Regions: A Case Study of Rossby Wave Breaking and Tropopause Folding**

Clemente Lopez-Bravo, Ernesto Caetano and Armenia Franco-Díaz

**General comments**

The manuscript describes the dynamic processes that led to a well-known ozone exchange process, i.e., STE, which primarily influences mid-tropospheric ozone and occasionally surface ozone, thereby impacting air quality. The STE event was thoroughly characterized using reanalysis data and Lagrangian trajectories. This case study is of great interest to the atmospheric science community; however, the manuscript requires major revisions in terms of its current structure and concise presentation of its results form before it can be accepted for publication in WCD.

I think sections 1.1 and 2 should be better organized by distinguishing clearly between subsections for data and method. Additionally, Section 1.1 presents some results before describing the methodology for obtaining those results, which could be remedied by reordering these sections.

The manuscript contains repetitive descriptions of the same event in multiple sections and subsections (3 and 4) of the manuscript. Some of the information described in these sections could be easily included in the introduction and discussed only when needed.

Although section 3.6 attempts to provide context, it feels out of place. In my opinion, this attempt to contextualize the STE event presents an opportunity to remedy one of the study's fundamental deficiencies, namely, a more thorough elaboration of its relevance (introduction), and to extend the study's period in a more systematic way (method), to later focus specifically on the 6-14 March 2016 event.

**Minor comment**

Lines 38-40: This line is unclear, linking geography and local meteorology with an independent variable: emissions. I suggest rewriting this sentence.

Line 53: Could you state the mixing ratio (by volume) value of this event?

Line 57: Satellite products are mentioned in the introduction as part of the analysis; however, the satellites used are not described in the methodology.

Line 59: Clarify the term "precursor" here and throughout the manuscript.

Line 89-91: For a general scientific community, I would not recommend using local indexes such as IMECA. The message would be the same and clearer if it just utilized nmol/mol (ppbv).

Line 91: The term "anomalies" is not clear. Do you mean high values, or was the ozone anomaly calculated?

Figure 2: Note that you are using mixing ratios ppb (I assume by volume), which is the same as nmol per mol, instead of concentration. Please correct throughout the manuscript.

Line 203-204: This is just an example of many sentences that can be moved to the introduction: "*Cut-off cyclones of this type have been argued to play an important role in*

*STE processes (Holton et al., 1995), as these structures can promote the irreversible descent of stratospheric air into the troposphere*".

Line 242: mass mixing ratios. Between what level pressures?

Line 281-289: If the authors want to include this text, it is first necessary to describe the methodology to estimate the tropospheric column ozone as well as to discuss the limitations of vertical resolution of satellite products, particularly in the UTLS. Consider that high elevation terrain also contributes to uncertainty. A full description of the tropopause estimation is also necessary to include when satellite products are utilized.

Line 326: Instead of "Phase I Mexican environmental contingency," can the author just state the mixing ratio values reached at the surface level?

Line 328: Temporal precursor? In another part of the manuscript, the terms "synoptic precursor," "synoptic dynamical precursor," and "precursor" are used interchangeably. Can the author define the term?

Line 330: Indicate the pressure levels where those ozone mixing ratios were identified.

---

## Referee Comment (RC2)

Review of **Stratosphere–Troposphere Exchange and Surface Ozone Pollution over Tropical Regions: A Case Study of Rossby Wave Breaking and Tropopause Folding**

by

Clemente Lopez-Bravo, Ernesto Caetano and Armenia Franco-Díaz

**Major Comments**

I agree with the first reviewer that this manuscript is not suitable for publication in the state that it in at the moment. It needs to be significantly reorganised and redundant information to be removed. This constitute a major revision of the manuscript and below I present some comments for the authors to consider.

In addition to the re-organisation of the manuscript, in my opinion as reviewer, this paper does not clearly illustrate what the contribution is to the STE body of knowledge thus far. It merely describes the event but how does this event advance our understand of STE processes? This is not clear to me.

**Minor comments**

Lines 28-29: The authors should clarify exactly how stratospheric air ends up in the tropopause. The occurrence of a RWB (or tropopause fold) event does not necessarily mean that STE occurs, what it guarantees is that the dynamical tropopause has been lowered. If no cut-off occurs so that a blob of stratospheric air is isolated in the troposphere below the tropopause, then how we would conclude that STE has occurred. If a tropopause lowers without a cut-off then stratospheric air remains above the dynamical tropopause. Please consider this issue as I find it to be consistently misunderstood right through the manuscript (as the comment on Fig 6 will illustrate this point).

General comment on the literature review in the introduction is that whilst it is well written, it does not go far in enough in reviewing STE issues and what this work attempts to close. Therefore, review of current STE knowledge needs to be significantly improved.

Lines 69 – 96: Section 1.1 does not belong to the introduction, where we normally present current knowledge, define hypothesis etc. It should be integrated into 3.1

Lines 97 – 159: The data and methods section should only discuss the diagnostics and defer discussion about the case to the results section. For instance the discussion of Figs 2 and 3 is treated here and it should be in Section 3.

Line 136 – 137: The authors are discussing breaking baroclinic waves and therefore should refer to potential vorticity conservation instead of absolute vorticity conservation, as the latter applied to barotropic Rossby waves. Also please consider the conditions under which PV is conserved and these should be mentioned explicitly here.

Lines 151-158: Should be moved to the results where the case is described. For this plot (Fig 3), please specify the range of longitudes through which the zonal averages are calculated for the two panels.

Sections 3.1 and 3.2. Some of the issues raised in 3.1 are repeated in 3.2, so these two sections should be combined and discussed using either Fig 4 or 5 but not both. My preference is the former and the streamers diagnostics could be implemented in Fig. 2. The geopotential heights and the appearance of the COL in the traditional sense has already been covered in Figure 1. Hoskins et al (1985) should also be invoked here as the PV anomaly induces the closed COL circulation

Lines 215 – 233: Fig 6 should show the evolution of the fold that corresponds to Fig 4, so that the time lag between the PV intrusion and the steep rise in ozone concentrations may be better explained. Also this discussion should show what was raised earlier in this review and that is, for there to be stratospheric exchange, a COL should have happened. The lat/pressure plots should clearly show this.

Another advantage of presenting the evolution of these zonal profiles in that we will have a 3-D view of the event, as readers.

Lines 275 – 278: Even though intuition is forcing me to agree with this statement. Fig 8 is not showing that the air that is associated with increased ozone concentrations and the tip of the curved arrow. Reason for this is that the PV values located near the surface are less than 2PVU, meaning that they cannot originate from the stratosphere. If anything, the air there originates from below the dynamical tropopause. The authors will have to explain.

Lines 352 – 354: The authors raise the issue of the role of topography without providing an analysis on these issues. In other words there is no analysis that suggest what the role of topography might be in this paper.

Conclusions: I am afraid I find that this study does not add to the body of literature to help us understand STE better in a convincing manner. It is highly descriptive in many areas and lacks a demonstration of how this case study helps us understand these issues. In addition to this, the paper requires very major restructuring. I am inclined towards rejection of the manuscript.